# Disentangling factors of variation in deep representations using adversarial training

**Michael Mathieu, Junbo Zhao, Pablo Sprechmann, Aditya Ramesh, Yann LeCun**
719 Broadway, 12th Floor, New York, NY 10003
{mathieu, junbo.zhao, pablo, ar2922, yann}@cs.nyu.edu

## Abstract

We introduce a conditional generative model for learning to disentangle the hidden factors of variation within a set of labeled observations, and separate them into complementary codes. One code summarizes the *specified* factors of variation associated with the labels. The other summarizes the remaining *unspecified* variability. During training, the only available source of supervision comes from our ability to distinguish among different observations belonging to the same class. Examples of such observations include images of a set of labeled objects captured at different viewpoints, or recordings of set of speakers dictating multiple phrases. In both instances, the intra-class diversity is the source of the unspecified factors of variation: each object is observed at multiple viewpoints, and each speaker dictates multiple phrases. Learning to disentangle the specified factors from the unspecified ones becomes easier when strong supervision is possible. Suppose that during training, we have access to pairs of images, where each pair shows two different objects captured from the same viewpoint. This source of alignment allows us to solve our task using existing methods. However, labels for the unspecified factors are usually unavailable in realistic scenarios where data acquisition is not strictly controlled. We address the problem of disentaglement in this more general setting by combining deep convolutional autoencoders with a form of adversarial training. Both factors of variation are implicitly captured in the organization of the learned embedding space, and can be used for solving single-image analogies. Experimental results on synthetic and real datasets show that the proposed method is capable of generalizing to unseen classes and intra-class variabilities.

## 1 Introduction

A fundamental challenge in understanding sensory data is learning to disentangle the underlying factors of variation that give rise to the observations [1]. For instance, the factors of variation involved in generating a speech recording include the speaker's attributes, such as gender, age, or accent, as well as the intonation and words being spoken. Similarly, the factors of variation underlying the image of an object include the object's physical representation and the viewing conditions. The difficulty of disentangling these hidden factors is that, in most real-world situations, each can influence the observation in a different and unpredictable way. It is seldom the case that one has access to rich forms of labeled data in which the nature of these influences is given explicitly.

Often times, the purpose for which a dataset is collected is to further progress in solving a certain supervised learning task. This type of learning is driven completely by the labels. The goal is for the learned representation to be invariant to factors of variation that are uninformative to the task at hand. While recent approaches for supervised learning have enjoyed tremendous success, their performance comes at the cost of discarding sources of variation that may be important for solving

other, closely-related tasks. Ideally, we would like to be able to learn representations in which the uninformative factors of variation are separated from the informative ones, instead of being discarded.

Many other exciting applications require the use of generative models that are capable of synthesizing novel instances where certain key factors of variation are held fixed. Unlike classification, generative modeling requires preserving all factors of variation. But merely preserving these factors is not sufficient for many tasks of interest, making the disentanglement process necessary. For example, in speech synthesis, one may wish to transfer one person's dialog to another person's voice. Inverse problems in image processing, such as denoising and super-resolution, require generating images that are perceptually consistent with corrupted or incomplete observations.

In this work, we introduce a deep conditional generative model that learns to separate the factors of variation associated with the labels from the other sources of variability. We only make the weak assumption that we are able to distinguish between observations assigned to the same label during training. To make disentanglement possible in this more general setting, we leverage both Variational Auto-Encoders (VAEs) [12, 25] and Generative Adversarial Networks (GANs) [9].

## 2   Related work

There is a vast literature on learning disentangled representations. Bilinear models [26] were an early approach to separate content and style for images of faces and text in various fonts. What-where autoencoders [22, 28] combine discrimination and reconstruction criteria to attempt to recover the factors of variation not associated with the labels. In [10], an autoencoder is trained to separate a translation invariant representation from a code that is used to recover the translation information. In [2], the authors show that standard deep architectures can discover and explicitly represent factors of variation aside those relevant for classification, by combining autoencoders with simple regularization terms during the training. In the context of generative models, the work in [23] extends the Restricted Boltzmann Machine by partitioning its hidden state into distinct factors of variation. The work presented in [11] uses a VAE in a semi-supervised learning setting. Their approach is able to disentangle the label information from the hidden code by providing an additional one-hot vector as input to the generative model. Similarly, [18] shows that autoencoders trained in a semi-supervised manner can transfer handwritten digit styles using a decoder conditioned on a categorical variable indicating the desired digit class. The main difference between these approaches and ours is that the former cannot generalize to unseen identities.

The work in [5, 13] further explores the application of content and style disentanglement to computer graphics. Whereas computer graphics involves going from an abstract description of a scene to a rendering, these methods learn to go backward from the rendering to recover the abstract description. This description can include attributes such as orientation and lighting information. While these methods are capable of producing impressive results, they benefit from being able to use synthetic data, making strong supervision possible.

Closely related to the problem of disentangling factors of variations in representation learning is that of learning fair representations [17, 7]. In particular, the Fair Variational Auto-Encoder [17] aims to learn representations that are invariant to certain nuisance factors of variation, while retaining as much of the remaining information as possible. The authors propose a variant of the VAE that encourages independence between the different latent factors of variation.

The problem of disentangling factors of variation also plays an important role in completing image analogies, the goal of the end-to-end model proposed in [24]. Their method relies on having access to matching examples during training. Our approach requires neither matching observations nor labels aside from the class identities. These properties allow the model to be trained on data with a large number of labels, enabling generalizing over the classes present in the training data.

## 3   Background

### 3.1   Variational autoencoder

The VAE framework is an approach for modeling a data distribution using a collection of independent latent variables. Let $x$ be a random variable (real or binary) representing the observed data and $z$ a collection of real-valued latent variables. The generative model over the pair $(x, z)$ is given by

$p(x, z) = p(x \mid z)p(z)$, where $p(z)$ is the prior distribution over the latent variables and $p(x \mid z)$ is the conditional likelihood function. Generally, we assume that the components of $z$ are independent Bernoulli or Gaussian random variables. The likelihood function is parameterized by a deep neural network referred to as the *decoder*.

A key aspect of VAEs is the use of a learned approximate inference procedure that is trained purely using gradient-based methods [12, 25]. This is achieved by using a learned approximate posterior $q(z \mid x) = N(\mu, \sigma I)$ whose parameters are given by another deep neural network referred to as the *encoder*. Thus, we have $z \sim \text{Enc}(x) = q(z|x)$ and $\tilde{x} \text{ Dec}(z) = p(x|z)$. The parameters of these networks are optimized by minimizing the upper-bound on the expected negative log-likelihood of $x$, which is given by

$$\mathbb{E}_{q(z \mid x)}[-\log p_\theta(x \mid z)] + \text{KL}(q(z|x) \parallel p(z)). \tag{1}$$

The first term in (1) corresponds to the reconstruction error, and the second term is a regularizer that ensures that the approximate posterior stays close to the prior.

### 3.2 Generative adversarial networks

Generative Adversarial Networks (GAN) [9] have enjoyed great success at producing realistic natural images [21]. The main idea is to use an auxiliary network Disc, called the *discriminator*, in conjunction with the generative model, Gen. The training procedure establishes a min-max game between the two networks as follows. On one hand, the discriminator is trained to differentiate between natural samples sampled from the true data distribution, and synthetic images produced by the generative model. On the other hand, the generator is trained to produce samples that confuse the discriminator into mistaking them for genuine images. The goal is for the generator to produce increasingly more realistic images as the discriminator learns to pick up on increasingly more subtle inaccuracies that allow it to tell apart real and fake images.

Both Disc and Gen can be conditioned on the label of the input that we wish to classify or generate, respectively [20]. This approach has been successfully used to produce samples that belong to a specific class or possess some desirable property [4, 19, 21]. The training objective can be expressed as a min-max problem given by

$$\min_{\text{Gen}} \max_{\text{Disc}} L_{\text{gan}}, \quad \text{where} \quad L_{\text{gan}} = \log \text{Disc}(x, \text{id}) + \log(1 - \text{Disc}(\text{Gen}(z, \text{id}), \text{id})). \tag{2}$$

where $p_d(x, id)$ is the data distribution conditioned on a given class label $id$, and $p(z)$ is a generic prior over the latent space (e.g. $N(0, I)$).

## 4 Model

### 4.1 Conditional generative model

We introduce a conditional probabilistic model admitting two independent sources of variation: an observed variable $s$ that characterizes the specified factors of variation, and a continuous latent variable $z$ that characterizes the remaining variability. The variable $s$ is given by a vector of real numbers, rather than a class ordinal or a one-hot vector, as we intend for the model to generalize to unseen identities.

Given an observed specified component $s$, we can sample

$$z \sim p(z) = N(0, I) \quad \text{and} \quad x \sim p_\theta(x \mid z, s), \tag{3}$$

in order to generate a new instance $x$ compatible with $s$.

The variables $s$ and $z$ are marginally independent, which promotes disentanglement between the specified and unspecified factors of variation. Again here, $p_\theta(x|z, s)$ is a likelihood function described by and decoder network, Dec, and the approximate posterior is modeled using an independent Gaussian distribution, $q_\phi(z|x, s) = N(\mu, \sigma I)$, whose parameters are specified via an encoder network, Enc. In this new setting, the variational upper-bound is be given by

$$\mathbb{E}_{q(z \mid x, s)}[-\log p_\theta(x \mid z, s)] + \text{KL}(q(z \mid x, s) \mid p(z)). \tag{4}$$

The specified component $s$ can be obtained from one or more images belonging to the same class. In this work, we consider the simplest case in which $s$ is obtained from a single image. To this end,

we define a deterministic encoder $f_s$ that maps images to their corresponding specified components. All sources of stochasticity in $s$ come from the data distribution. The conditional likelihood given by (3) can now be written as $x \sim p_\theta(x \mid z, f_s(x'))$ where $x'$ is any image sharing the same label as $x$, including $x$ itself. In addition to $f_s$, the model has an additional encoder $f_z$ that parameterizes the approximate posterior $q(z \mid x, s)$. It is natural to consider an architecture in which parameters of both encoders are shared.

We now define a single encoder Enc by $\text{Enc}(x) = (f_s(x), f_z(x)) = (s, (\mu, \sigma) = (s, z)$, where $s$ is the specified component, and $z = (\mu, \sigma)$ the parameters of the approximate posterior that constitute the unspecified component. To generate a new instance, we synthesize $s$ and $z$ using Dec to obtain $\tilde{x} = \text{Dec}(s, z)$.

The model described above cannot be trained by minimizing the log-likelihood alone. In particular, there is nothing that prevents all of the information about the observation from flowing through the unspecified component. The decoder could learn to ignore $s$, and the approximate posterior could map images belonging to the same class to different regions of the latent space. This degenerate solution can be easily prevented when we have access to labels for the unspecified factors of variation, as in [24]. In this case, we could enforce that $s$ be informative by requiring that Dec be able to reconstruct two observations having the same unspecified label after their unspecified components are swapped. But for many real-world scenarios, it is either impractical or impossible to obtain labels for the unspecified factors of variation. In the following section, we explain a way of eliminating the need for such labels.

## 4.2 Discriminative regularization

An alternative approach to preventing the degenerate solution described in the previous section, without the need for labels for the unspecified components, makes use of GANs (3.2). As before, we employ a procedure in which the unspecified components of a pair of observations are swapped. But since the observations need not be aligned along the unspecified factors of variation, it no longer makes sense to enforce reconstruction. After swapping, the class identities of both observations will remain the same, but the sources of variability within their corresponding classes will change. Hence, rather than enforcing reconstruction, we ensure that both observations are assigned high probabilities of belonging to their original classes by an external discriminator. Formally, we introduce the discriminative term given by (2) into the loss given by (5), yielding

$$\mathbb{E}_{q(z \mid x,s)}[-\log p_\theta(x \mid z, s)] + \text{KL}(q(z \mid x, s) \mid\mid p(z)) + \lambda L_{\text{gan}}, \tag{5}$$

where $\lambda$ is a non-negative weight.

Recent works have explored combining VAE with GAN [14, 6]. These approaches aim at including a recognition network (allowing solving inference problems) to the GAN framework. In the setting used in this work, GAN is used to compensate the lack of aligned training data. The work in [14] investigates the use of GANs for obtaining perceptually better loss functions (beyond pixels). While this is not the goal of our work, our framework is able to generate sharper images, which comes as a side effect. We evaluated including a GAN loss also for samples, however, the system became unstable without leading to perceptually better generations. An interesting variant could be to use separate discriminator for images generated with and without supervision.

## 4.3 Training procedure

Let $x_1$ and $x_1'$ be samples sharing the same label, namely $id_1$, and $x_2$ a sample belonging to a different class, $id_2$. On one hand we want to minimize the upper bound of negative log likelihood of $x_1$ when feeding to the decoder inputs of the form $(z_1, f_s(x_1))$ and $(z_1, f_s(x_1'))$, where $z_1$ are samples form the approximate posterior $q(z|x_1)$. On the other hand, we want to minimize the adversarial loss of samples generated by feeding to the decoder inputs given by $(z, f_s(x_2))$, where $z$ is sampled from the approximate posterior $q(z|x_1)$. This corresponds to swapping specified and unspecified factors of $x_1$ and $x_2$. We could only use upper bound if we had access to aligned data. As in the GAN setting described in Section 3.2, we alternate this procedure with updates of the adversary network. The diagram of the network is shown in figure 1, and the described training procedure is summarized in on Algorithm 1, in the supplementary material.

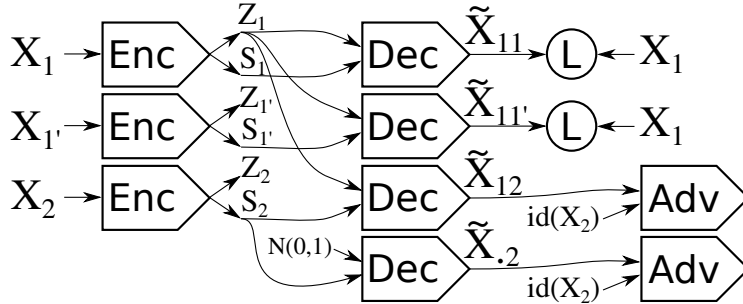

Figure 1: Training architecture. The inputs $x_1$ and $x_1'$ are two different samples with the same label, whereas $x_2$ can have any label.

## 5 Experiments

**Datasets.** We evaluate our model on both synthetic and real datasets: Sprites dataset [24], MNIST [15], NORB [16] and the Extended-YaleB dataset [8]. We used Torch7 [3] to conduct all experiments. The network architectures follow that of DCGAN [21] and are described in detail in the supplementary material.

**Evaluation.** To the best of our knowledge, there is no standard benchmark dataset (or task) for evaluating disentangling performance [2]. We propose two forms of evaluation to illustrate the behavior of the proposed framework, one qualitative and one quantitative.

Qualitative evaluation is obtained by visually examining the perceptual quality of single-image analogies and conditional images generation. For all datasets, we evaluated the models in four different settings: *swapping:* given a pair of images, we generate samples conditioning on the specified component extracted from one of the images and sampling from the approximate posterior obtained from the other one. This procedure is analogous to the sampling technique employed during training, described in Section 4.3, and corresponds to solving single-image analogies; *retrieval:* in order to asses the correlation between the specified and unspecified components, we performed nearest neighbor retrieval in the learned embedding spaces. We computed the corresponding representations for all samples (for the unspecified component we used the mean of the approximate posterior distribution) and then retrieved the nearest neighbors for a given query image; *interpolation:* to evaluate the coverage of the data manifold, we generated a sequence of images by linearly interpolating the codes of two given test images (for both specified and unspecified representations); *conditional generation:* given a test image, we generate samples conditioning on its specified component, sampling directly from the prior distribution, $p(z)$. In all the experiments images were randomly chosen from the test set, please see specific details for each dataset.

The objective evaluation of generative models is a difficult task and itself subject of current research [27]. Frequent evaluation metrics, such as measuring the log-likelihood of a set of validation samples, are often not very meaningful as they do not correlate to the perceptual quality of the images [27]. Furthermore, the loss function used by our model does not correspond a bound on the likelihood of a generative model, which would render this evaluation less meaningful. As a quantitative measure, we evaluate the degree of disentanglement via a classification task. Namely, we measure how much information about the identity is contained in the specified and unspecified components.

**MNIST.** In this setup, the specified part is simply the class of the digit. The goal is to show that the model is able to learn to disentangle the style from the identity of the digit and to produce satisfactory analogies. We cannot test the ability of the model to generalize to unseen identities. In this case, one could directly condition on a class label [11, 18]. It is still interesting that the proposed model is able to transfer handwriting style without having access to matched examples while still be able to learn a smooth representation of the digits as show in the interpolation results. Results are shown in Figure 2. We observe that the generated images are convincing and particularly sharp, the latter is an "side-effect" produced by the GAN term in our training loss.

**Sprites.** The dataset is composed of 672 unique characters (we refer to them as sprites), each of which is associated with 20 animations [24]. Any image of a sprite can present 7 sources of variation: body type, gender, hair type, armor type, arm type, greaves type, and weapon type. Unlike the work in [24], we do not use any supervision regarding the positions of the sprites. The results obtained for

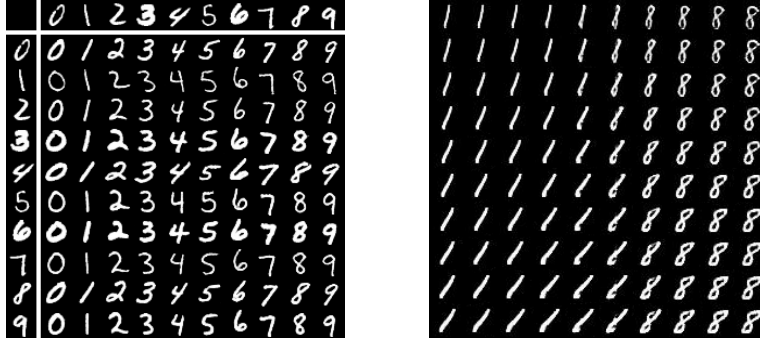

Figure 2: left(a): A visualization grid of 2D MNIST image swapping generation. The top row and leftmost column digits come from the test set. The other digits are generated using $z$ from leftmost digit, and $s$ from the digit at the top of the column. The diagonal digits show reconstructions. Right(b): Interpolation visualization. Digits located at top-left corner and bottom-right corner come from the dataset. The rest digits are generated by interpolating $s$ and $z$. Like (a), each row has constant a $z$ each column a constant $s$.

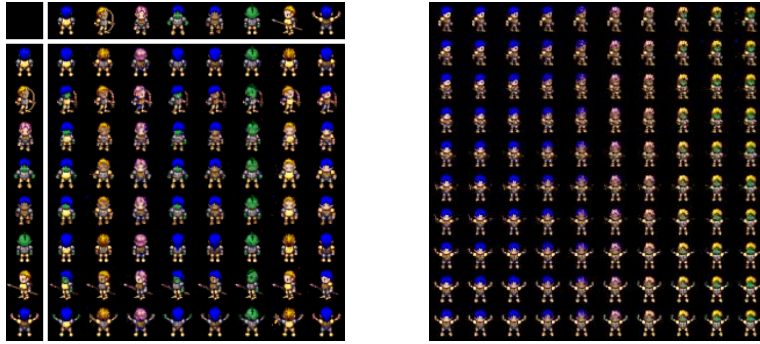

Figure 3: left(a): A visualization grid of 2D sprites swapping generation. Same visualization arrangement as in 2(a); right(b): Interpolation visualization. Same arrangement as in 2(b).

the swapping and interpolation settings are displayed in Figure 3 while retrieval result are showed in 4. Samples from the conditional model are shown in 5(a). We observe that the model is able to generalize to unseen sprites quite well. The generated images are sharp and single image analogies are resolved successfully. The interpolation results show that one can smoothly transition between identities or positions. It is worth noting that this dataset has a fixed number of discrete positions. Thus, 3(b) shows a reasonable coverage of the manifold with some abrupt changes. For instance, the hands are not moving up from the pixel space, but appearing gradually from the faint background.

**NORB.** For the NORB dataset we used instance identity (rather than object category) for defining the labels. This results in 25 different object identities in the training set and another 25 distinct objects identities in the testing set. As in the sprite dataset, the identities used at testing have never been presented to the network at training time. In this case, however, the small number of identities seen at training time makes the generalization more difficult. In Figure 6 we present results for interpolation and swapping. We observe that the model is able to resolve analogies well. However, the quality of the results are degraded. In particular, classes having high variability (such as planes) are not reconstructed well. Also some of the models are highly symmetric, thus creating a lot of uncertainty. We conjecture that these problems could be eliminated in the presence of more training data. Queries in the case of NORB are not as expressive as with the sprites, but we can still observe good behavior. We refer to these images to the supplementary material.

**Extended-YaleB.** The datasets consists of facial images of 28 individuals taken under different positions and illuminations. The training and testing sets contains roughly 600 and 180 images per individual respectively. Figure 7 shows interpolation and swapping results for a set of testing images. Due to the small number of identities, we cannot test in this case the generalization to unseen identities. We observe that the model is able to resolve the analogies in a satisfactory, position and illumination are transferred correctly although these positions have not been seen at train time for

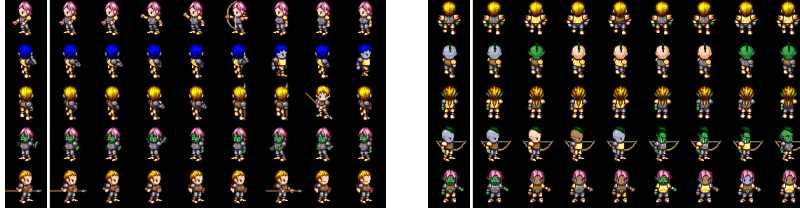

Figure 4: left(a): sprite retrieval querying on *specified* component; right(b): sprite retrieval querying on unspecified component. Sprites placed at the left of the white lane are used as the query.

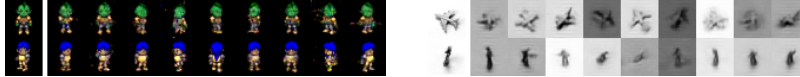

Figure 5: left(a): sprite generation by sampling; right(b): NORB generation by sampling.

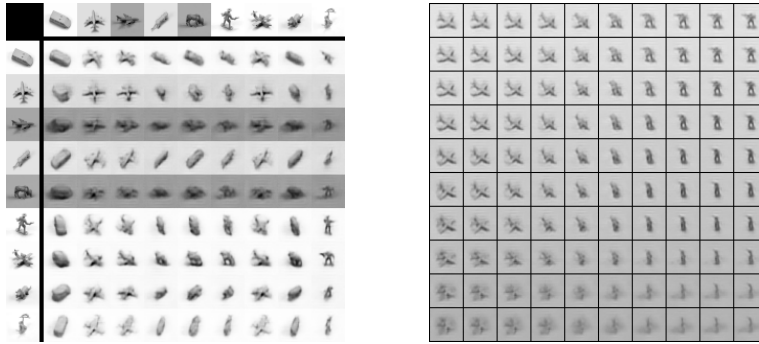

Figure 6: left(a): A visualization grid of 2D NORB image swapping generation. Same visualization arrangement as in 2(a); right(b): Interpolation visualization. Same arrangement as in 2(b).

these individuals. In the supplementary material we show samples drawn from the conditional model as well as other examples of interpolation and swapping.

**Quantitative evaluation.** We analyze the disentanglement of the specified and unspecified representations, by using them as input features for a prediction task. We trained a two-layer neural network with 256 hidden units to predict structured labels for the sprite dataset, toy category for the NORB dataset (four-legged animals, human figures, airplanes, trucks, and cars) and the subject identity for Extended-YaleB dataset. We used early-stopping on a validation set to prevent overfitting. We report both training and testing errors in Table 1. In all cases the unspecified component is agnostic to the identity information, almost matching the performance of random selection. On the other hand, the specified components are highly informative, producing almost the same results as a classifier directly trained on a discriminative manner. In particular, we observe some overfitting in the NORB dataset. This might also be due to the difficulty of generalizing to unseen identities using a small dataset.

**Influence of components of the framework**. It is worth evaluating the contribution of the different components of the framework. Without the adversarial regularization, the model is unable to learn disentangled representations. It can be verified empirically that the unspecified component is completely ignored, as discussed in Section 4.1. A valid question to ask is if the training of $s$ has be done jointly in an end-to-end manner or could be pre-computed. In Section 4 of the supplementary material we run our setting by using an embedding trained before hand to classify the identities. The model is still able to learned a disentangled representations. The quality of the generated images as well as the analogies are compromised. Better pre-trained embeddings could be considered, for example, enforcing the representation of different images to be close to each other and far from those corresponding to different identities. However, joint end-to-end training has still the advantage of requiring fewer parameters, due to the parameter sharing of the encoders.

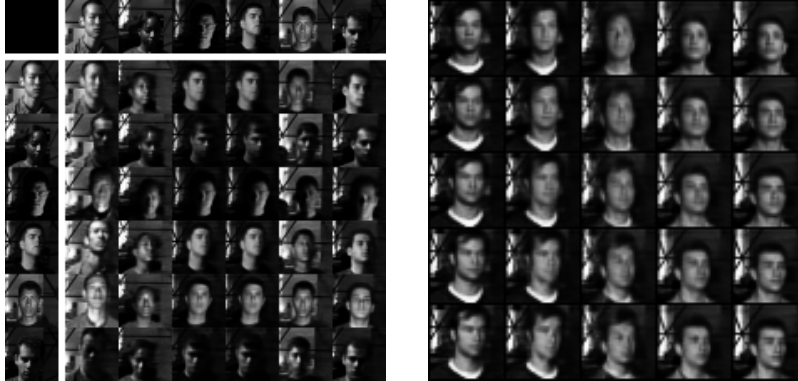

Figure 7: left(a): A visualization grid of 2D Extended-YaleB face image swapping generation. right(b): Interpolation visualization. See 2 for description.

Table 1: Comparison of classification upon $z$ and $s$. Shown numbers are all error rate.

| set | Sprites | | NORB | | Extended-YaleB | |
|---|---|---|---|---|---|---|
| | $z$ | $s$ | $z$ | $s$ | $z$ | $s$ |
| train | 58.6% | 5.5% | 79.8% | 2.6% | 96.4% | 0.05% |
| test | 59.8% | 5.2% | 79.9% | 13.5% | 96.4% | 0.08% |
| random-chance | 60.7% | | 80.0% | | 96.4% | |

## 6 Conclusions and discussion

This paper presents a conditional generative model that learns to disentangle the factors of variations of the data specified and unspecified through a given categorization. The proposed model does not rely on strong supervision regarding the sources of variations. This is achieved by combining two very successful generative models: VAE and GAN. The model is able to resolve the analogies in a consistent way on several datasets with minimal parameter/architecture tuning. Although this initial results are promising there is a lot to be tested and understood. The model is motivated on a general settings that is expected to encounter in more realistic scenarios. However, in this initial study we only tested the model on rather constrained examples. As was observed in the results shown using the NORB dataset, given the weaker supervision assumed in our setting, the proposed approach seems to have a high sample complexity relying on training samples covering the full range of variations for both specified and unspecified variations. The proposed model does not attempt to disentangle variations within the specified and unspecified components. There are many possible ways of mapping a unit Gaussian to corresponding images, in the current setting, there is nothing preventing the obtained mapping to present highly entangled factors of variations.

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
