[Supplementary Material · disentangling-factors-variation-supplementary.pdf]

# Supplementary material: Disentangling factors of variation in deep representations using adversarial training

**Michael Mathieu, Junbo Zhao, Aditya Ramesh, Pablo Sprechmann, Yann LeCun**
719 Broadway, 12th Floor, New York, NY 10003
{mathieu, junbo.zhao, pablo, yann}@cs.nyu.edu

## 1   Network architectures

The encoder consists of a shared sub-network that splits into two separate branches. In our experiments with MNIST and the Sprites datasets, the shared sub-network is composed by three 5x5 convolutional layers with stride 2, using spatial batch normalization (BN) **?** and ReLU non-linearities. For the NORB and YaleB datasets, we use six 3x3 convolutional layers, with stride 2 every other layer. The output from the top convolution layer is split into two sub-networks. One parametrizes the approximate posterior of the unspecified component and consists of a fully-connected (FC) layer, producing two outputs corresponding to mean and variance of the approximate posterior (modeling the unspecified component). The other sub-network is also a fully connected used to produce the $s$ vector modeling the specified component. The decoder network takes a sample $z$ and a vector $s$ as inputs. Both codes go through a fully connected network. These representations are merged together by directly adding them and fed into a feed-forward network composed by a network mirroring encoder structure (replacing the strides by fractional strides). The discriminator is conditioned on the label, $id$, and configured following that used in (conditional) DCGAN. It contains three 5x5 convolutional layers with stride 2, using BN and Leaky-ReLU with slope 0.2. The label goes through three independent lookup tables and are added at the three first layers of representation. The dimensionality of each representation varies from dataset to dataset. They were obtained by monitoring the results on a validation set. For MNIST, we used 16 coefficients for each component. For sprites, NORB and Extended-YaleB, we set their dimensions as 64 and 512 for specified and unspecified components respectively. We found that using Stochastic Gradient Descent gives good results.

## 2   Image generation

Figure 8 shows image generation. The specified part is extracted from a data sample, and an unspecified part is sampled from a Gaussian distribution. The generated sample show variation within the category of the specified part.

## 3   Interpolation

Figure 9 shows more interpolation results. The specified and unspecified parts are extracted from two images are interpolated independently.

## 4   Using a pre-trained embedding

In order to access the advantage of jointly training the system to learn the specified and unspecified parts, we tried another training scheme, summarized in the following two-step approach:

Figure 8: More image generation. The specified part is extracted from the left images, and the unspecified part is sampled to generate the images on the right-hand side.

Figure 9: Interpolation figures, on the yaleB dataset. Only the top-left and bottom-right real faces from the test set. The the lines show interpolation along the specified part and the column show interpolation along the unspecified part.

- Add a two-layer neural network on top of the specified part of the encoder, followed by a classification loss. Train this system in a plain supervised fashion to learn the class of the samples. When the system is converged, freeze the weights.
- Add another encoder to produce the unspecified part of the code, and train the system as before (keeping the weights of the specified encoder frozen).

Figure 10 show the generation grid swapping the specified and unspecified parts (similar to figure 2a).

Figure 10: Swapping grid of the specified and unspecified part (see figure 2a for more details). (a) Left: pre-training the specified part of the encoder on a purely supervised task (b) Right: jointly training the whole system.

# 5 Training procedure

Algorithm 1 summarizes the whole training procedure. The notations are defined in sections 3 and 4 of the main paper.

---
**Algorithm 1** Full model training

---
**for** number of training iterations **do**

    *Train the generative model*

    Sample a triplet of samples $(x_1, id_1)$, $(x'_1, id_1)$, $(x_2, id_2)$ where $x_1$ and $x'_1$ have the same label

    Compute the codes $(\mu_1, \sigma_1, s_1) = \text{Enc}(x_1)$, $(\mu'_1, \sigma'_1, s'_1) = \text{Enc}(x'_1)$, $(\mu_2, \sigma_2, s_2) = \text{Enc}(x_2)$

    Sample $z_1 \sim N(\mu_1, \sigma_1)$, $z'_1 \sim N(\mu'_1, \sigma'_1)$, $z_2 \sim N(\mu_2, \sigma_2)$

    Compute the reconstructions $\tilde{X_{11}} = \text{Dec}(z_1, s_1)$, $\quad \tilde{X_{11'}} = \text{Dec}(z_1, s'_1)$

    Compute the loss between $\tilde{X_{11}}$ and $X_1$, and between $\tilde{X_{11'}}$ and $X_1$, and backpropagate the gradients

    Compute the generation $\tilde{X_{12}} = \text{Dec}(z_1, s_2)$ and the adversarial loss $\log(\text{Adv}(\tilde{X_{12}}, id_2))$, and backpropagate the gradients, keeping the weights of Adv frozen

    Sample $z \sim N(0, 1)$, generate $X_{\cdot 2} = \text{Dec}(z, s_2)$, compute the adversarial loss $\log(\text{Adv}(\tilde{X_{\cdot 2}}, id_2))$ and backpropagate the gradients, keeping the weights of Adv frozen

    *Train the adversary*

    Sample a pair of samples $(x_1, id_1)$, $(x_2, id_2)$

    Compute the codes $(\mu_1, \sigma_1, s_1) = \text{Enc}(x_1)$, $\quad (\mu_2, \sigma_2, s_2) = \text{Enc}(x_2)$

    Sample $z_1 \sim N(\mu_1, \sigma_1)$, $z_2 \sim N(\mu_2, \sigma_2)$

    Compute the reconstructions $\tilde{X_{11}} = \text{Dec}(z_1, s_1)$, $\quad \tilde{X_{12}} = \text{Dec}(z_1, s_2)$

    Compute the adversarial loss (negative sample) $\log(1 - \text{Adv}(X_{12}, id_2))$ and backpropagate the gradients, keeping the weights of Enc and Dec frozen

    Compute the adversarial loss (positive sample) $\log(\text{Adv}(X_2, id_2))$ and backpropagate the gradients, keeping the weights of Enc and Dec frozen

**end for**

---