[Reviews · NeurIPS 2016]

Reviewer 1

Summary

The authors presented a new generative model that learns to disentangle the factors of variations of the data. The authors claim that the proposed model is pretty robust to supervision. This is achieved by combining two of the most successful generative models: VAE and GAN. The model is able to resolve the analogies in a consistent way on several datasets with minimal parameter/architecture tunning.

Qualitative Assessment

The experimental results are very weak (non-existent). Based on this, I believe this work is very preliminary and is not is a position to be published in NIPS. I would encourage the authors to do a more thorough experimental study.

Confidence in this Review

2-Confident (read it all; understood it all reasonably well)


Reviewer 2

Summary

This paper presents a way to learn latent codes for data, that captures both the information relevant for a given classification task, as well as the remaining irrelevant factors of variation (rather than discarding the latter as a classification model would). This is done by combining a VAE-style generative model, and adversarial training. This model proves capable of disentangling style and content in images (without explicit supervision for style information), and proves useful for analogy resolution.

Qualitative Assessment

This is a paper presenting a significantly useful and novel technique, and doing so with a great deal of attention to detail and clear presentation. The experimental section might need improvement. The authors might want to put up a stronger argumentation as to why learning to disentangle relevant and irrelevant information (with regarded to a supervised task) is useful and important, v.s. discarding irrelevant information altogether. It might be obvious to the authors but might not be self-evident for people coming from a classification background. Currently this is justified by stating that it is "a more satisfying approach", with no real justification given. Clearly citing what the practical applications are would make a better point. Section 3 and in particular 3.3 is somewhat confusing (although the paper is clear and well-explained overall). The experimental section should do a better job at presenting clearly what hypothesis the authors are trying to prove, and how the results do prove it. Implicitly it seems that the unstated hypotheses vary from experiment to experiment, and it isn't always clear what the presented results are supposed to show. The experimental section currently reads along the line of "here's what we did, here's what we got", which is detrimental to the overall quality of the paper. Despite these issues, overall it's definitely a good paper with interesting results.

Confidence in this Review

2-Confident (read it all; understood it all reasonably well)


Reviewer 3

Summary

This paper introduces a generative model for learning to disentangle hidden factors of variation. The disentangling separates the code into two, where one is claimed to be the code that descries factors relevant to solving a specific task, and the other describing the remaining factors. Experimental results show that the proposed method is promising.

Qualitative Assessment

The fact that disentangling does not depend on strong supervision of the particular sources of variation makes this approach powerful and practically very relevant. To me, the experiments presented on various different datasets and tasks are satisfactory to show that this approach might be a promising direction.

Confidence in this Review

2-Confident (read it all; understood it all reasonably well)


Reviewer 4

Summary

The authors combine state of the art methods VAE and GAN to generate images with two complementary codes: one relevant and one irrelevant. They major contribution of the paper is the development of a training procedure that exploits triplets of images (two sharing the relevant code, one note sharing) to regularize the encoder-decoder architecture and avoid trivial solutions. The results are qualitatively good and comparable to previous article using more sources of supervision.

Qualitative Assessment

The article visibly integrate a detailed exploration of the most recent generative deep-learning literature. The overall model is composed by a combination of multiple recently introduced loss functions, often differently from the original papers, conditioned on class labels. It is quite difficult to understand the exact contribution to the final qualitative result of each of the components. Since so much care is used into avoiding trivial solution between Z and S it would be nice to understand how much shared the parameters can be, and how much care is necessary to train the models on different dataset. The limitation of the paper are given by lengthy description of the model and the fact that the triplet type of training is not extensively explained, but seems to be the fundamental contribution improving on the necessity of having a label for each factor of variation of the previously published literature. It would be interesting to see novel experiments enabled by the contribution of the paper. On the overall I think the paper deserves to be accepted for a poster contribution as it properly integrates state of the art results with an interesting training procedure enabling similar performance with reduced sources of supervision.

Confidence in this Review

2-Confident (read it all; understood it all reasonably well)


Reviewer 5

Summary

This paper proposes a method to separate style and content in a neural representation. This is achieved by defining a latent code with vectors for specified ("content") and unspecified ("style") variation. A GAN loss encourages samples from the unspecified vector conditioned on the specified vector to be indistinguishable from empirical samples from a specified content class. This way the unspecified vector is encouraged to model all variation not related to class. Unlike previous methods, this is achieved without forcing the latent code to explicitly include a categorical class variable. Experiments show that style and content are successfully separated on a few simple datasets.

Qualitative Assessment

I found this to be a quite interesting paper. It is well-written, except for many typos throughout, which are easily fixable. I did, however, find the model somewhat hard to follow. I think it would help to clarify early on what is the difference between using s in the latent code versus using a class label indicator. This distinction strikes me as a primary contribution of the paper. The model is novel to my knowledge, and uses GANs in an intriguing way. The experiments are convincing that style and content were successfully separated. I think the main weakness of this paper is that the experiments do not convincingly show that the novelty of the model paid off. A primary novelty is that the proposed model does not require class labels in the code layer. Instead the code layer encodes class-specific information as a continuous vector of features. The paper argues that an advantage might be that the model can be applied to novel classes, not seen at training time. Unfortunately, the experiments show little evidence that this actually works. The NORB experiments are the only ones that test performance on novel classes, and in these examples it does not look like style is well separated from content. As a result, it is not clear what is gained by coding class-specific features s rather than using the older approach of coding a discrete class label in the latent code. The paper does, however, show that using s allows for smooth interpolation between classes, which is an interesting property. Overall this paper has demonstrated a novel method to disentangle factors of variation, employing a GAN, and this method has potential advantages in terms of how it codes class-specific information (and that it doesn't rely on aligned training data). I think the paper could be improved by including more tests of generalization to novel classes (surely this works on MNIST?), and perhaps finding other ways to exploit the novel encoding of class-specific information.

Confidence in this Review

2-Confident (read it all; understood it all reasonably well)


Reviewer 6

Summary

Paper seeks to explore the variations amongst samples which separate multiple classes using auto encoders and decoders. Specifically, the authors propose combining generative adversarial networks and variational auto encoders. The idea mimics the game play between two opponents, where one attempts to fool the other into believing a synthetic sample is in fact a natural sample. The paper proposes an iterative training procedure where a generative model was first trained on a number of samples while keeping the weights of the adversary constant and later the adversary is trained while keeping the generative model weights constant. The paper performs experiments on generation of instances between classes, retrieval of instances belonging to a given class, and interpolation of instances between two classes. The experiments were performed on MNIST, a set of 2D character animation sprites, and 2D NORB toy image dataset.

Qualitative Assessment

The author(s) seek to explore the factors of variations which exist amongst the samples within and between classes, but their work instead proposes a framework that expresses the variations amongst classes. The paper failed to provide a qualitative analysis about such factors, but instead takes an almost black box approach that provides "pretty" results. Perhaps reasoning and backing of how such variations can be extracted/utilized would be helpful in bridging your work to wider subject areas. The author(s) stress the originality of their framework; however, putting together two well known frameworks is common in vision tasks and the basis of the idea for their training procedure is analogous to that of the famous AlphaGo. Although the results are visually appealing, the MNIST and 2D NORB datasets in which the experiments were performed on is widely considered to be toy datasets (I am unfamiliar with the sprites dataset). The authors may consider including experiments on more difficult dataset (ie. evaluating a mixture of multiple expressions in Face dataset). In addition, the paper should be checked for spelling, typos and grammatical errors as some of such error caused a number of confusions during the review process.

Confidence in this Review

2-Confident (read it all; understood it all reasonably well)